# Antimicrobial Resistance in *Escherichia coli* and Its Correlation with Antimicrobial Use on Commercial Poultry Farms in Bangladesh

**DOI:** 10.3390/antibiotics12091361

**Published:** 2023-08-24

**Authors:** Nelima Ibrahim, Filip Boyen, Md. Abu Shoieb Mohsin, Moniek Ringenier, Anna Catharina Berge, Ilias Chantziaras, Guillaume Fournié, Dirk Pfeiffer, Jeroen Dewulf

**Affiliations:** 1Faculty of Veterinary Medicine, Department of Internal Medicine, Reproduction and Population Medicine, Ghent University, Salisburylaan 133, 9820 Merelbeke, Belgium; 2Department of Livestock Services, Dhaka 1215, Bangladesh; 3Faculty of Veterinary Medicine, Department of Pathobiology, Pharmacology and Zoological Medicine, Ghent University, Salisburylaan 133, 9820 Merelbeke, Belgium; 4Department of Medicine and Surgery, Chattogram Veterinary and Animal Sciences University, Chattogram 4202, Bangladesh; shoieb.tox@gmail.com; 5Université de Lyon, INRAE, VetAgro Sup, UMR EPIA, 69280 Marcy l’Etoile, France; 6Université Clermont Auvergne, INRAE, VetAgro Sup, UMR EPIA, 63122 Saint Genes Champanelle, France; 7Department of Pathobiology and Population Sciences, Royal Veterinary College, Royal College Street, London NW1 0TU, UK; 8Department of Infectious Diseases and Public Health, Jockey Club College of Veterinary Medicine and Life Sciences, City University of Hong Kong, Hong Kong SAR, China

**Keywords:** MIC, fecal and environmental samples, ECOFF value, CLSI, broiler, Sonali, multidrug resistance (MDR), *E. coli*, Bangladesh

## Abstract

Antimicrobial resistance is a global concern, posing risks to human and animal health. This research quantified antimicrobial resistance (AMR) in *E. coli* isolates from poultry fecal and environmental samples in Bangladesh and explored their association with antimicrobial use (AMU). We screened 725 fecal and 250 environmental samples from 94 conventional broilers and 51 Sonali farms for *E. coli* presence using MALDI-TOF mass spectrometry. AMU data were collected at flock levels, expressed as treatment incidence (TI), while minimum inhibitory concentrations (MIC) for 14 antibiotics were determined on five fecal *E. coli* isolates per farm and on all environmental isolates. MIC results were interpreted using human clinical breakpoints and EUCAST epidemiological cut-off values (ECOFFs). Acquired resistance against commonly used antimicrobial agents such as ciprofloxacin, tetracycline and ampicillin, was extremely high and predominantly clinically relevant. There was a moderate correlation between fecal and environmental antibiotic resistance index (ARI), but there was no significant correlation between AMU and AMR, suggesting that the observed AMR prevalence is unrelated to current AMU in poultry, but may be due to high historical AMU. A high level of multidrug resistance, including against critically important antimicrobials, was found in both farm types. Therefore, an AMR/AMU surveillance program is urgently needed in the poultry production sector of Bangladesh.

## 1. Introduction 

Antimicrobial resistance (AMR) has been identified by the World Health Organization (WHO) as one of the major threats to public health in the present and near future [1]. The WHO launched a Global Action Plan (GAP) in 2015 to tackle this developing global issue in a comprehensive manner, based on a ‘One Health’ strategy that emphasizes the interdependence of human, animal and environmental health [2,3]. South Asia is considered to be exposed to the highest risk of AMR among all WHO regions due to its large human population and high level of antimicrobial use (AMU) in both humans and animals [4]. The escalation of evolutionary trends leading to AMR poses a significant threat to the health of both humans and animals [2,5,6]. Apart from their crucial role in treating and preventing human infections, antibiotics have been extensively used in food-producing animals. However, this practice poses a significant concern as it creates a reservoir of antibiotic-resistant bacteria and AMR genes, which have the potential to be transferred to humans [7]. This transfer of antibiotic resistance from animals to humans further exacerbates the global challenge of antimicrobial resistance and brings attention to the effectiveness of antibiotics in human healthcare [6,8,9]. The World Organization for Animal Health (WOAH) recommends monitoring AMR in commensal *E. coli* sampled from animals [10]. Indeed, although commensal *E. coli* is known not to be harmful to their host, several studies have shown that *E. coli* can develop resistance and serve as a reservoir for multidrug resistance (MDR) both in animal populations and environment, making it a useful indicator organism for measuring antimicrobial resistance (AMR) [11,12,13]. According to the Bangladesh Fish Feed and Animal Feed Act, 2010, the government banned the use of antimicrobials in animal feed [14]. In accordance with WHO GAP standards, Bangladesh has approved a National Action Plan (NAP) to combat AMR [15]. Population density, easy access and inappropriate use of antimicrobials, contamination of environments with animal manure, all contribute to the occurrence of drug-resistant community-acquired infections [16,17,18,19,20]. In Bangladesh, Sonali chicken consumption holds the second position after broiler meat in terms of popularity [18]. Sonali chicken, which is a cross-breed of Rhode Island Red cocks and Fayoumi hens, exhibits a similar appearance and taste to native chickens. In Bangladesh, several studies have been conducted on the prevalence of MDR *E. coli* in broilers [21,22,23,24,25]. However, none of these studies have investigated the link between AMR and AMU. Additionally, there is a lack of available data on AMR of *E. coli* in Sonali chickens. Furthermore, there is a notable absence of studies that use epidemiological cut-off values (ECOFFs) to interpret the results. ECOFF values provide a standardized framework for interpreting antimicrobial susceptibility data, allowing for more reliable comparisons and informed decision-making. To address these gaps, this study aimed to quantify antimicrobial resistance patterns in *E. coli* isolates obtained from chicken fecal and environmental samples collected from farms raising conventional broiler and Sonali chickens, the most produced and consumed poultry types in Bangladesh [18], using appropriate testing protocols and interpretation criteria. In addition, the associations between antimicrobial use (AMU) and resistance (AMR), as well as any differences in resistance among broiler and Sonali chicken isolates, were explored.

## 2. Results

In total, 725 fresh fecal and 250 environmental samples were collected. *E. coli* was recovered in 98% and 78% of fecal and environmental samples, respectively. The number of isolates obtained per farm ranged from three to seven. Minimum inhibitory concentrations (MICs) testing was conducted on 691 fecal and 191 environmental *E. coli* isolates, corresponding to 3–5 isolates randomly selected per farm. The MIC values for QC strains were within the acceptable ranges as described by CLSI [26]. 

### 2.1. AMR Results

Results of the antimicrobial susceptibility testing (MIC distributions) are summarized in Table 1 for fecal and in Table 2 for environmental isolates of both broiler and Sonali farms. The Appendix A provide separate results for broiler and Sonali farms, including findings for both fecal and environmental samples (Appendix A). All fecal and environmental isolates showed MDR towards three to seven antibiotic classes. Based on ECOFFs, the proportion of isolates resistant (i.e., non-wild type) to ciprofloxacin, tetracycline, ampicillin, nalidixic acid, trimethoprim or sulfamethaxole ranged from 93% to 99%. About two-thirds of fecal isolates were resistant to chloramphenicol or azithromycin. In contrast, about two-thirds of isolates were wild types with respect to gentamicin or tigecycline and the proportion of resistant isolates was ≤6% for ceftazidime and cefotaxime, whereas for colistin, it was 12%. The antimicrobial compounds associated with the highest levels of resistance in the environmental isolates were the same as in fecal samples, despite slightly lower proportion of resistant isolates. The main difference was for chloramphenicol, with the proportion of resistant isolates dropping to a third of all tested environmental isolates. No significant difference was observed in the resistance prevalence between broiler and Sonali fecal and environmental isolates. 

Acquired resistance in non-wild-type isolates was predominantly clinically relevant as assessed using CLSI clinical breakpoints (Table 1 and Table 2). 

As shown in Figure 1, a significant (*p* < 0.001) correlation (R^2^ = 0.35) was found between the antimicrobial resistance index (ARI) of fecal and environmental isolates in both broiler and Sonali.

### 2.2. AMU in Flock Level

Antimicrobials were used on all farms, but there was a huge variation in the amount of AMU in the different farms. The quantification of AMU (TI_DDDvet_) and the detailed results were described by Ibrahim et al., 2023 [27]. The median of treatment incidence which expresses the number of Defined Daily Dose (TI_DDDvet_) was 60.0 (range 18.3–188.2) for conventional broilers and 58.3 (range 31.1–212.6) for Sonali chickens. This indicates that conventional broilers and Sonali birds were treated with antimicrobials for approximately 60% and 58% of their lifetime, respectively [27]. 

### 2.3. Linking Antimicrobial Use and Antimicrobial Resistance

Prior to exploring the association between antimicrobial use and antimicrobial resistance, Figure 2 depicts the frequency distribution of the ARI. 

There was no significant (*p* = 0.73) correlation between flock level of total antimicrobial use (TI_DDDvet_) and antimicrobial resistance (ARI), as shown in Figure 3. 

In both broiler and Sonali farms, ciprofloxacin, ampicillin, tetracycline, colistin, gentamicin and trimethoprim were reported as being most commonly used. In Table 3, the association between the use of these molecules and the observed resistance prevalence is shown. Only for tetracycline (*p* = 0.01) and gentamicin (*p* = 0.02) a significant association was found between the use of the compound in the farm and the resistance prevalence.

## 3. Discussion

The emergence of antimicrobial resistance has become a major animal and public health threat. The current findings reveal a high level of antimicrobial resistance in fecal and environmental isolates towards the most commonly used antimicrobials in both broiler and Sonali farms. Furthermore, in all fecal and environmental isolates, 100% of the investigated poultry *E. coli* strains showed MDR towards three to seven antibiotic classes. Although this is not the first report of MDR *E. coli* in poultry in Bangladesh [21,22,23,24,25,28,29] this manuscript’s results have added value over previous reports both in terms of the methods used and interpretation of the results. For example, we have used MIC testing, compared to disk diffusion in the past, which is generally regarded as less reliable than MIC testing and is not reliable at all for certain antibiotics, such as colistin [30] and we interpreted susceptibility testing results using both wild-type cut-off value and clinical breakpoints. Additionally, this manuscript describes for the first time, the association between antimicrobial usage (AMU) and antimicrobial resistance (AMR), as well as the correlation between ARI of fecal and environmental isolates based on Bangladesh data. 

The *E. coli* isolates obtained both from fecal and environmental samples from broiler and Sonali chicken in the current study show acquired resistance in varying degrees to different antimicrobial agents. All the isolates collected from fecal samples in this study had similar to higher resistance levels (higher percentage of resistant bacteria) to commonly used antimicrobials on the farm compared to some of the previous studies in Bangladesh [21,22,23,24,25]. Based on most recent publications [28,31], *E. coli* in poultry and poultry environments were found to have varying but generally high degrees of ampicillin, ciprofloxacin, tetracycline, sulfamethoxazole and trimethoprim resistance in Bangladesh, reaching up to 100%, similar to this study. These high levels of acquired resistance are likely due to the long-term use of these antimicrobials in poultry in Bangladesh. 

In the current study, colistin resistance was found in 12% of fecal isolates, which may more closely estimate colistin resistance prevalence in poultry in Bangladesh, compared with previous investigations using the disk diffusion method, which reported higher resistance rates [21,22,28], that is not reliable for colistin susceptibility testing [21,30]. Since many laboratories still rely on the cheaper disk diffusion test, the emergence of colistin resistance may be misjudged and needs to be monitored closely using the appropriate test methods [30]. Nevertheless, it has been shown using molecular methods that mobilized colistin resistance (mcr) genes, which are associated with colistin resistance, has been detected in up to 25% of *E. coli* isolates obtained from poultry in Bangladesh [32]. It is important to mention that during the time of this study, antimicrobial combinations including colistin, but excluding the liquid single oral solution in bottles of at least one liter, were prohibited by the Directorate General of Drug Administration in 2019 [33]. However, despite this ban, farmers continued to use colistin due to its availability in liquid form on the market. Finally, colistin was fully prohibited in all forms in 2022 [34]. It is unclear to what extent the observed lower resistance prevalence for colistin in the current study, compared to previous studies in Bangladesh, is due to the use of a different methodology of sampling and/or resistance detection or due to the (partial) ban of colistin use. Further molecular characterization of current isolates may confirm observed phenotypic results but were out of the scope of the current investigations. 

In case of cefotaxime, this study found 5% and 7% resistance in fecal and environmental isolates, respectively, whereas high resistance has been reported in isolates from poultry cloacal swabs and farm sewage samples in Bangladesh, though in a different region and using a different susceptibility testing protocol [22]. In this study, *E. coli* isolates obtained from both fecal and environmental samples were 100% wild type for meropenem, though other studies showed meropenem resistance in cloacal samples, sewerage and hand washes samples in Bangladesh [22,23,35]. Considering the fact that meropenem is probably not used in poultry in Bangladesh, the rare occasion of isolating meropenem-resistant *E. coli* in poultry, likely indicates human to poultry transmission of such isolates. 

In this study, *E. coli* isolates demonstrated high levels of acquired resistance (98% and 89% in fecal and environmental isolates, respectively) to the quinolone-class antibiotic ciprofloxacin [36]. Ciprofloxacin has been widely used in commercial poultry farms in Bangladesh over the last decade [33,35] though its use in poultry is strictly regulated in the European Union (EU) or even forbidden in the USA and in large parts of the world [37,38]. According to a recent study in Bangladesh, fluoroquinolones were the most frequently used antimicrobial class in broiler chickens [27]. In this context, combinations of ciprofloxacin with trimethoprim were banned by the Directorate General of Drug Administration in Bangladesh in 2019 [33]. Furthermore, ciprofloxacin use can cause cross-resistance to other members of the quinolone class [36]. Consequently, resistance to nalidixic acid was found to be 96% and 72% for fecal and environmental isolates, respectively, despite the fact that this antibiotic was not used in the farms of Bangladesh. 

Azithromycin is a commonly used macrolide for the treatment of invasive *E. coli* infections in humans in Bangladesh [28]. The fact that 65% fecal and 67% environmental isolates obtained in the current study had acquired resistance against this critically important agent indicates a potential serious human health issue. The high percentage of azithromycin resistance in *E. coli* isolates found in poultry and poultry environments in the current study is somewhat unexpected because this antibiotic is not commonly used in the poultry farms of Bangladesh [27]. One explanation might be the fact that farmers often raise other animals such as cattle or goats on the same farm and azithromycin is a commonly used antibiotic for large animal treatment in Bangladesh [29]. In a recent study, Amin et al. (2020) found that *E. coli* resistance against azithromycin was 100% in cattle and their environment [29]. In this context, azithromycin was recently banned for veterinary use by the Directorate General of Drug Administration, Bangladesh (2022) [34].

We also interpreted the MIC results using human clinical breakpoint showing that the acquired resistance was very often also clinically relevant. For example, in Bangladesh, fluoroquinolones, which are considered as a first-line antibiotic therapy for *E. coli* infections [28], are widely used to treat bacterial infections in humans, poultry and other animals. 

The fecal AMR index and environmental AMR index had a moderate correlation of 0.54. This correlation indicates that flocks contaminated the farm environment and/or vice versa. This could be attributed to the lack of farm biosecurity and the fact that poultry litter is often dried on the farm premises before sale, or directly used as a fertilizer in vegetable fields. This correlation, however, was only moderate, implying that there could be other (fecal) sources of *E. coli* isolates with a different antimicrobial resistance pattern in the farm environment. Possible sources might be the proximity of other farms or animal species (cattle, goat, etc.), or even the presence of humans and related wastewater. Resistance in environmental isolates may also be indirectly related to AMU in chickens and other host species through resistance selection induced by antibiotic residues present in both manure and wastewater. Contact with these bacteria has the potential to spread AMR in humans. 

All the *E. coli* isolates (100%) of the current study showed MDR against at least three, but up to seven antimicrobial classes which is consistent with previous studies [25,39,40,41]. According to a recent comprehensive analysis, food animals and particularly poultry, are probably responsible for a proportion of *E. coli* infections in humans with extra-intestinal, extended-spectrum cephalosporin resistance [41,42]. Interventions that restrict antibiotic use in food-producing animals have been linked to a decrease in the presence of antibiotic-resistant bacteria in these animals [43]. There is a limited but indicative set of evidence that suggests a similar link in the human populations studied, particularly those with direct exposure to food-producing animals [43]. 

In the current study, the observed associations between AMU and AMR were very weak to absent. Stuart Levy introduced the threshold theory, which suggests that a certain level of antimicrobial drug consumption is required to trigger the emergence of resistance in a particular environment [44]. Austin et al. (1999) supported this theory by describing the sigmoidal rise in resistance over time in the presence of a constant rate of antimicrobial consumption [45]. This suggests that small changes in the amount of antimicrobials used in a population with a low level of AMR may lead to much larger changes in resistance than the effect of comparable changes in use in a population where already a (very) high level of resistance is present [46]. This might explain why we could not find significant associations. It also highlights the importance of reacting on emerging resistance at the earliest possible phase. Furthermore, when studying the link between use and resistance in bacteria using field data, the observed levels of resistance are a reflection of current and historical use, whereas measured use often only reflects recent use or, at best, a retrospect of only a short period [36,47,48,49]. 

The Bangladesh government has developed a National Action Plan (NAP) for the period 2017–2022 to combat AMR [15]. The main goals were to identify and restrict the sale of critical antimicrobials used in food animals, to stop the “over-the-counter drug sale”, to monitor and assess compliance with withdrawal periods, by providing training to farmers and poultry workers to raise awareness. However, the findings of this study clearly indicate that these goals have not been achieved so far. Establishing a regulatory framework (e.g., antimicrobial use law) to control AMU is of utmost importance in Bangladesh to effectively combat AMR. Given that the current timeframe of the NAP has expired, it is essential to extend it and identify the reasons for the NAP’s shortcomings in order to take appropriate measures for its successful implementation. The findings of this study are expected to provide valuable insights for policymakers and practitioners, aiding them in revising the NAP to address the present circumstances and expedite its practical execution. To comprehend the resistance mechanisms and relatedness of the *E. coli* strains, future research should involve conducting genotypic resistance and phylogenetic analysis due to the high level of resistance observed.

## 4. Materials and Methods

### 4.1. Study Design

A cross-sectional study was conducted in seven (7) districts of Bangladesh to collect AMU data and samples for isolating *E. coli*. A total of 145 small-scale (range: 500–2500 number of birds) commercial conventional broiler (94) and Sonali (51) poultry farms were recruited. The recruitment criteria of the farms were described in Ibrahim et al., 2023 [27]. Eligible farms adopted an all-in all-out production system and had a farm size of >1000 birds per batch. In cases where multiple sheds were present, only one shed was considered randomly. Each farm was visited twice, upon the delivery of day-old chicks and within two days prior to the chicken being sold [27]. 

On each farm, samples were collected on the second visit. Fresh feces were collected from 5 healthy-appearing chickens and environmental samples (swab from soil) were collected in the area between the farm gate and the shed and near the vegetable field due to the practice of using poultry litter as manure for vegetables. Two environmental samples were collected from each farm located in the northern districts (n = 105) and one environmental sample from each farm located in the southeast districts (n = 40). The swabs were transported in ice-pack cooled boxes to the Central Disease Investigation Laboratory (CDIL), Bangladesh where they were stored at –80 °C until further use.

Fecal samples were inoculated on MacConkey III agar (Oxoid company, Dhaka, Bangladesh) and incubated aerobically at 37 °C for 18 to 24 h. Environmental samples were first resuscitated in 10 mL Brain Heart Infusion (BHI) broth and incubated aerobically at 37 °C for 18 to 24 h before being inoculated on MacConkey agar and incubated aerobically at 37 °C for another 18 to 24 h. Before sending all lactose positive Enterobacteriaceae isolates to the laboratory of Ghent University, Belgium, these isolates were passaged three times on MacConkey III agar, as required by Belgian law to minimize the chance of importing Newcastle Disease virus and highly pathogenic Avian Influenza virus. 

Upon arrival at the laboratory, isolates were purified on Columbia agar with 5% sheep blood and subsequently confirmed to be *E. coli* using MALDI-TOF mass spectrometry, as described previously [50]. Antimicrobial susceptibility testing of the *E. coli* isolates was performed using a Sensititre EU Surveillance *E. coli* EUVSEC Plate (Trek Diagnostic Systems, Thermofisher Scientific, Merelbeke, Belgium) according to the manufacturer’s guidelines. In short, 1 to 3 colonies were suspended in sterile saline to an optical density of 0.5 McFarland. Fifty microliter of this suspension was inoculated in 10 mL sterile Mueller Hinton broth. Again, fifty microliter of the Mueller Hinton broth with bacteria was transferred to each of the wells in the Sensititre micro-titer plate with the lyophilized antimicrobials using a multichannel (final concentration of 2.5 × 10^4^ CFU/well).

After incubation at 35 °C for 18–24 h, the plates were examined and growth end-points were established for each antimicrobial to provide MICs. The Minimum inhibitory concentration (MIC) was defined as the lowest concentration by which no visible growth could be detected. Quality control (QC) strains, *E. coli* ATCC 25922 and *E. coli* NCTC 13846 (for QC colistin resistance), were used throughout the study [26]. MIC values were interpreted based on (1) the epidemiological cut-off values (ECOFFs) published by the European Committee on Antimicrobial Susceptibility Testing (EUCAST) [51] and (2) the human clinical breakpoints published by CLSI [26] as no clinical breakpoints for poultry were available (Table 4).

Isolates having MIC values greater or equal than the ECOFF were considered to have acquired resistance and classified as non-wild type. Isolates having MIC values greater than the clinical breakpoints for susceptibility or resistance were considered intermediate or resistant, respectively. Note that no clinical breakpoints were available for tigecycline and while no ECOFF value was available for sulfamethoxazole, non-wild type was assumed if a bi- or multimodal MIC distribution was observed.

The antimicrobial susceptibility results were primarily interpreted using the EUCAST epidemiological cut-off values [52], which identify whether an isolate has acquired resistance against a certain antibiotic compared to the wild-type population [53,54]. On the other hand, human clinical breakpoints provides insights into whether the observed acquired resistance patterns are clinically relevant. We therefore chose to report both interpretations.

The European Food Safety Authority (EFSA) defined antimicrobial resistance level as the percentage of tested isolates of a given microorganism that were resistant to a given antimicrobial. These levels are described as rare (<0.1%), very low (0.1% to 1%), low (>1% to 10%), moderate (>10% to 20%), high (>20% to 50%), very high (>50% to 70%) and extremely high (>70%) [55].

An isolate was defined as multi-drug resistant if it was resistant to antimicrobial compounds belonging to at least three different antimicrobial classes. The antimicrobial resistance index (ARI) of an isolate was calculated as the proportion of tested antimicrobial compounds against which resistance was observed. It was computed based on 13 rather than 14 compounds because cefotaxime and ceftazidime belong to the same antibiotic class and exhibited resistance simultaneously. The average antimicrobial resistance index (ARI) for each farm was calculated by determining the ARI of all isolates from that farm and taking the mean of those values.

### 4.2. AMU

The quantification of AMU was described in detail in Ibrahim et. al, 2023 [27]. In brief, AMU was quantified by computing the treatment incidence (TI) which expresses the number of Defined Daily Dose (DDDvet) administered per 100 animal days at risk. It reflects the percentage of the lifetime of a bird for which it was treated with antimicrobials.
$${TI}_{DDDvet}=\frac{total\;amount\;of\;AS\;administered\;or\;purchased}{DDDvet\;(mg/kg/day)\times no.\;of\;days\;at\;risk\times kg\;of\;AAR}\times 100\;AAR$$
AS—active substance; AAR—animal at risk.

### 4.3. Data Analysis

Descriptive statistics related to AMU (TI_DDDvet_) and AMR were computed using Excel^®^ 2016. Correlations between all variables were explored by means of Pearson’s correlation test. Additionally, R^2^ was computed to assess the proportion of variance in antimicrobial resistance explained by the TI. The assumptions of normality and homogeneity of variance were assessed by examining histograms and normal probability plots of residuals. A t-test was conducted to compare the resistance of *E. coli* between broiler and Sonali poultry. To test whether the use of antibiotics was associated with the presence of resistant *E. coli* strains, we used the generalized linear mixed model (GLMM) with a binomial logistic probability function. Farm was included as subject, isolate as within-subject factor and their working correlation were set as independent. The use or no use of antibiotic was included as fixed factor. Associations were considered significant when *p*-values were ≤0.05. The data were analyzed using SPSS software (SPSS version 27^®^; IBM, Armonk, NY, USA).

## 5. Conclusions

In Bangladesh, *E. coli* isolates obtained from poultry feces and the environment of broiler and Sonali chicken farms exhibit high levels of multidrug resistance to commonly used antibiotic classes including fluoroquinolones, which are classified as a “high priority critically important antibiotic” for humans. Unexpectedly, high antimicrobial use was not associated with the level of AMR, probably due to an overall (very) high level of resistance. The fecal and environmental AMR indexes were moderately correlated, which may indicate a lack of biosecurity on Bangladesh poultry farms. A comprehensive and multisectoral approach is necessary to address these factors and combat the spread of resistance in the poultry industry.

## Figures and Tables

**Figure 1 antibiotics-12-01361-f001:**
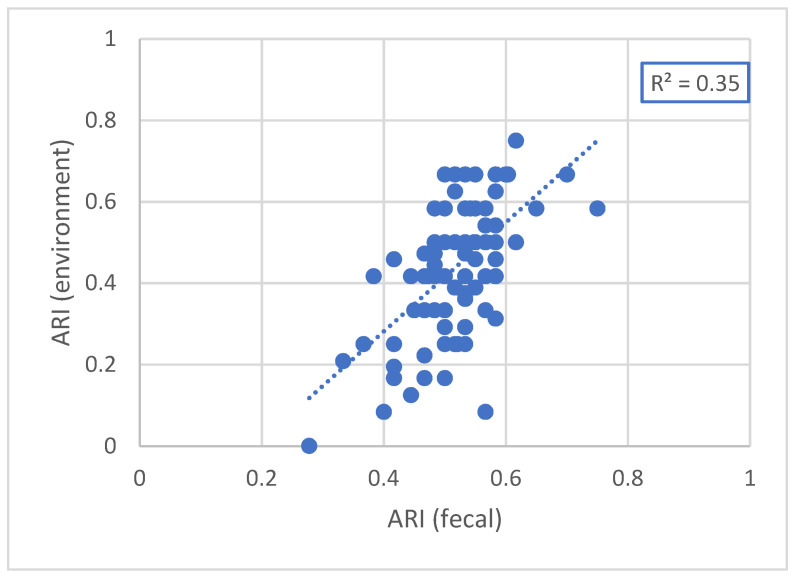
The correlation between antimicrobial resistance (ARI) of fecal and environmental isolates of all sampled farms.

**Figure 2 antibiotics-12-01361-f002:**
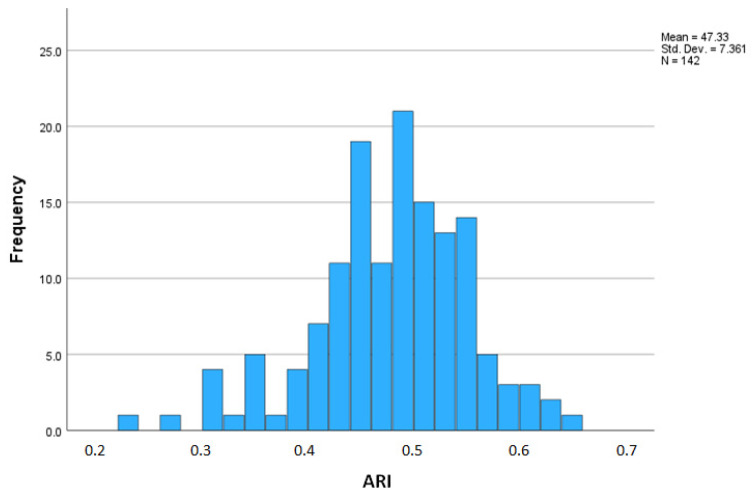
Distribution of antimicrobial resistance index of farms.

**Figure 3 antibiotics-12-01361-f003:**
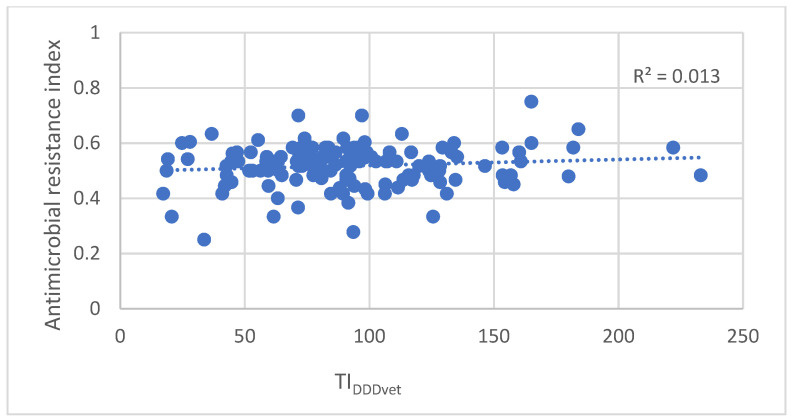
The correlation between antimicrobial use (TI_DDDvet_) and antimicrobial resistance index (ARI) on all sampled farms.

**Table 1 antibiotics-12-01361-t001:**
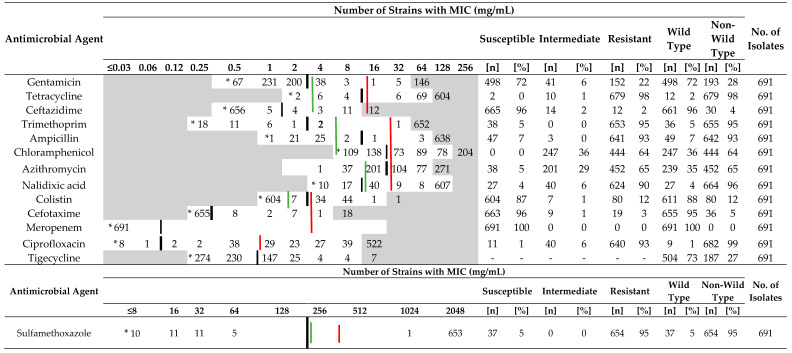
Minimum inhibitory concentration distribution for *E. coli* in fecal isolates.

* No visible growth at this concentration, meaning MIC is equal to or below this concentration. 

—CLSI breakpoint between susceptible and intermediate, 

—CLSI breakpoint between intermediate and resistant and 

—ECOFF value between wild type and non-wild type.

**Table 2 antibiotics-12-01361-t002:**
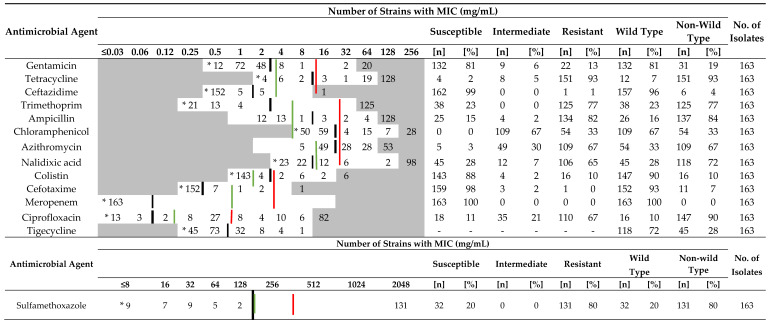
Minimum inhibitory concentration distribution for *E. coli* in environmental isolates.

* No visible growth at this concentration, meaning MIC is equal to or below this concentration. 

—CLSI breakpoint between susceptible and intermediate, 

—CLSI breakpoint between intermediate and resistant and 

— ECOFF value between wild type and non-wild type.

**Table 3 antibiotics-12-01361-t003:** Proportion of farms that used most common six (6) antimicrobials and two antimicrobial class.

Name of Antibiotics		Number of Farms	No. of Susceptible Isolates	No. of Resistant Isolates	Prevalence of Resistance (%)	*p* Value
Ciprofloxacin	Use	64	4	311	99	0.399
	No use	78	8	368	98	
Ampicillin	Use	91	33	422	93	0.983
	No use	51	17	219	93	
Tetracycline	Use	75	2	361	99	0.01
	No use	67	10	318	97	
Trimethoprim	Use	22	7	101	94	0.760
	No use	120	32	551	95	
Colistin	Use	80	378	30	7	0.570
	No use	62	266	17	6	
Gentamicin	Use	8	25	15	38	0.02
	No use	134	511	140	22	
Fluoroquinolones class	Use	89	9	485	98	0.812
	No use	53	3	194	98	
Sulfonamides class	Use	101	26	475	95	0.829
	No use	41	11	179	94	

**Table 4 antibiotics-12-01361-t004:** Panel of antimicrobial substances and concentration ranges included in antimicrobial susceptibility testing and applied epidemiological cut-offs (ECOFFs) and clinical breakpoints.

Antimicrobial (Abbreviation)	Concentration Range Tested (mg/L)	Non-Wild-Type Population * (mg/L)	Clinical Breakpoint for Susceptibility ^#^ (mg/L)	Clinical Breakpoint for Resistance ^#^ (mg/L)
Ampicillin (AMP)	1–64	≥8	≤8	≥32
Cefotaxime (FOT)	0.25–4	≥0.25	≤1	≥4
Ceftazidime (TAZ)	0.5–8	≥1	≤8	≥16
Meropenem (MERO)	0.03–16	≥0.06	≤1	≥4
Nalidixic acid (NAL)	4–128	≥8	-	≥32
Ciprofloxacin (CIP)	0.015–8	≥0.06	≤0.25	≥1
Tetracycline (TET)	2–64	≥8	≤4	≥16
Colistin (COL)	1–16	≥2	≤1	≥4
Gentamicin (GEN)	0.5–32	≥2	≤4	≥16
Trimethoprim (TMP)	0.25–32	≥2	-	≥16
Sulfamethoxazole (SMX)	8–1024	-	-	≥512
Chloramphenicol (CHL)	8–128	≥16	≤8	≥32
Azithromycin (AZI)	2–64	≥16	-	≥32
Tigecycline (TGC)	0.25–8	≥0.5	-	-

* According to EUCAST (https://mic.eucast.org/search/, most recently viewed on 15 May 2023); # According to CLSI M100-ED32:2022 [26].

## Data Availability

Data were provided in the manuscript.

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
