# Peer review of "Antimicrobial Resistance in Escherichia coli and Its Correlation with Antimicrobial Use on Commercial Poultry Farms in Bangladesh"

_antibiotics, 2023, doi:10.3390/antibiotics12091361_

Round 1
Reviewer 1 Report
Overview and general recommendation:
In this study, Authors evaluated the antimicrobial resistance of E. coli isolated from poultry faecal and environmental samples in Bangladesh and their association with antimicrobial use. This study is well conceived, conduced, and written. I have mainly minor comments/suggestions for the Authors, below included to improve the description throughout the manuscript.
Major comments:
- One concept conflicts with this study: at lines 65-66, Authors declared that “In Bangladesh, the use of antimicrobials in food producing animals has been banned” but, in the following paragraphs (i.e., in par.2.2, 2.3, or at line 417), Authors evaluated the antimicrobial use in poultry farms. I suggest to be clearer with this potential descriptive conflict.
Minor comments:
- Line 27: I suggest to evaluate if “evaluated” could be preferred to “explored”.
- Line 34: it is not clear the mean of “often clinically relevant”. Please, explain and, if necessary, revise it.
- Line 38: I suggest to add a comma after “antimicrobials”.
- Keywords: I suggest to reconsider some keywords, which could be replaced by more useful ones (i.e., E.coli or Bangladesh)
- Line 54: I suggest to replace “are” with “have been”, since the “extensive” use in food-producing animals is step by step less common worldwide.
- Line 74: “(MDR)” could be removed.
- Lines 83-84: “,the most produced…[18],” could be removed because redundant.
- Line 88: “samples” after “fecal” could be removed.
- Tables 1 and 2: it is not sufficiently clear the mean of vertical green and red lines withing the MIC values, please explain it, also in footnotes if necessary.
- Par.2.2: it is not clear if these results were obtained from this study, according to the methodology described by [27] (as expressed in par.4.2), or were duplicated by the study [27].
- Line 174: I suggest to include in brackets the p values considered of significant association.
- Line 182: Is the “current findings” referred to the results of this study?
- Line 183: “of” could be replaced by “toward”
- Line 186: I suggest to include the related reference after “Bangladesh”
- Line 203: I suggest to move the references at line 200, after “publications”.
- Line 213: I suggest to replace “mcr” with “mobilized colistin resistance (mcr)”
- Line 315: I suggest to include a numeric range to elucidate the mean of “small-scale” farms.
- Line 322: I suggest to briefly include some details on environmental sample types, since some suddenly appear at line 231.
- Line 344: hyphen in “Mueller-Hinton” could be removed.
- Table 4: almost all vertical and horizontal lines in this table could be removed.
- Lines 383-384: please, check this sentence, particularly for the use of terms “belong” and “exhibited”.
Reviewer 2 Report

The quality of English in my view is satisfactory
Reviewer 3 Report
The manuscript entitled “Antimicrobial resistance in Escherichia coli and correlation with antimicrobial use on commercial poultry farms in Bangladesh” describes the analysis of antimicrobial resistance in Escherichia coli isolated from poultry fecal and environmental samples in Bangladesh. It is a very high-concern topic from a “One Health” perspective.
As regards the language, a minor revision should be carried out to make the work more fluent and easier to read.
I think that this paper should be accepted for publication in “Antibiotics”, after some revisions;
In more detail:
Line 113: specify the “ARI” acronym meaning when first appearing in the text.
Line 140: explain the “TIDDDvet” acronym.
Line 147: ARI acronym has been already explained in the text, so, authors should write only “ARI”.
Line 283: insert “that” between evidence and suggests.
Line 335: To minimize THE chance…
Line 343: Fifty microliters of….were inoculated…Again “Fifty microliters were…”
Line 349: authors should already have explained the “MIC” acronym when first appearing in the text.
Please correct all the acronyms throughout the text. Explain them when first appearing in the text and then use only the acronym.
Line 390-396: please rephrase the sentence and write the formula more clearly.
Minor revisions should be carried out to make the work more fluent and easier to read
Reviewer 4 Report
The manuscript titled “Antimicrobial resistance in Escherichia coli and correlation with antimicrobial use on commercial poultry farms in Bangladesh” presents compelling findings regarding authors’ research on the global issue of antibiotic resistance in Bangladesh.
In their study, the Authors also investigate the correlation between antimicrobial resistance and antimicrobial use.
Notably, despite most of the relevant research papers investigate the AMR using the disk diffusion method, in this manuscript the authors employ the Minimal inhibitory concentration method that it is more reliable especially for some antibiotics.
Minor
Line 89: Can you explain why you haven’t isolated E.coli from the 2% of the collected fecal samples? That finding is really interesting as E.coli is present in the intestines in large numbers/gr
Line 321: “…feces were collected…” instead of “…feces was collected…”
Line 324: Is there any specific reason for not collecting the same number of environmental samples from the farms from the northern and southern districts?
Reviewer 5 Report
Dear authors,
you presented a research on AMR of E. coli and AMU on two different poultry production - broilers and Sonali chickens.
Introduction is well written, as well as Discussion and Conclusion.
In Results, brief description of MDR results of isolated E. coli can be included.
Specific comments:
L62 - "that the bacteria" - needs revision
L 113- here is ARI index mentioned for the first time, but the whole name (Antimicrobial Resistance Index) is mentioned in full in L146; as ARI means Antimicrobial Resistance Index, is there a need to write ARI index? No need for word "index" after ARI?
Table 3 - in the title you mention 6 antibiotics, and in the table there are 6 mentioned, as well as two groups?
L244-246 sentence needs revision
L 256 E. coli not in italic
L 329-336 please mention the manufacturer of the agars
L 335- Newcastle diseases (there is a space- New castle)
